# *IDH* Mutations in Chondrosarcoma: Case Closed or Not?

**DOI:** 10.3390/cancers15143603

**Published:** 2023-07-13

**Authors:** Sanne Venneker, Judith V. M. G. Bovée

**Affiliations:** Department of Pathology, Leiden University Medical Center, 2333 ZA Leiden, The Netherlands

**Keywords:** sarcoma, chondrosarcoma, isocitrate dehydrogenase mutation, *IDH1*, *IDH2*, D-2-hydroxyglutarate

## Abstract

**Simple Summary:**

Chondrosarcomas are cartilage tumours that often harbour a mutation in one of the isocitrate dehydrogenase (*IDH*) genes. *IDH* mutations are important drivers at the beginning of cartilage tumour development, but their role in later stages remains unclear. However, other *IDH* mutant tumour types do show an influence of this mutation on patient outcomes and therapies that specifically kill these *IDH* mutant tumour cells. Factors that could explain this discrepancy in the role of *IDH* mutations are differences in tumour type, elevated oncometabolite levels, the type of model used in preclinical studies (natural vs. introduced *IDH* mutation), and additional (epi)genetic alterations. The latter influence the downstream biological effects of an *IDH* mutation, and recent studies have indeed identified subgroups within *IDH* wildtype and mutant chondrosarcomas. Future studies should build upon these subgroups to improve the identification of effective treatments and biomarkers that predict which patients will benefit from these therapies.

**Abstract:**

Chondrosarcomas are malignant cartilage-producing tumours that frequently harbour isocitrate dehydrogenase 1 and -2 (*IDH*) gene mutations. Several studies have confirmed that these mutations are key players in the early stages of cartilage tumour development, but their role in later stages remains ambiguous. The prognostic value of *IDH* mutations remains unclear and preclinical studies have not identified effective treatment modalities (in)directly targeting these mutations. In contrast, the *IDH* mutation status is a prognostic factor in other cancers, and IDH mutant inhibitors as well as therapeutic strategies targeting the underlying vulnerabilities induced by *IDH* mutations seem effective in these tumour types. This discrepancy in findings might be ascribed to a difference in tumour type, elevated D-2-hydroxyglutarate levels, and the type of in vitro model (endogenous vs. genetically modified) used in preclinical studies. Moreover, recent studies suggest that the (epi)genetic landscape in which the *IDH* mutation functions is an important factor to consider when investigating potential therapeutic strategies or patient outcomes. These findings imply that the dichotomy between *IDH* wildtype and mutant is too simplistic and additional subgroups indeed exist within chondrosarcoma. Future studies should focus on the identification, characterisation, and tailoring of treatments towards these biological subgroups within *IDH* wildtype and mutant chondrosarcoma.

## 1. Introduction

Chondrosarcomas are malignant cartilage-producing tumours that account for 20% of all malignant bone tumours [1,2]. Enchondromas are considered the benign precursor lesions of chondrosarcoma, but progression towards malignant tumours is rarely seen (<1%) outside the non-hereditary syndromes (i.e., Ollier disease and Maffucci syndrome) that cause multiple cartilaginous neoplasms (enchondromatosis) [3,4]. Chondrosarcomas arise predominantly in the third to sixth decades of life and can affect the long as well as the flat bones, especially the femur, humerus, pelvis, and ribs, and occasionally the spine or base of the skull. Pathological characteristics divide chondrosarcoma into several subtypes, including conventional chondrosarcoma (85%), dedifferentiated chondrosarcoma (10%), and rare subtypes that include mesenchymal, clear cell, and periosteal chondrosarcoma (5%). Based on the anatomical location, conventional chondrosarcoma can be further subdivided into central (i.e., in the medulla of the bone) and peripheral (i.e., at the surface of the bone) conventional chondrosarcoma (85% and 15%, respectively) [1,2]. 

Histological grading is defined by, among other factors, the mitotic count, the presence of spindle-shaped cells, cellularity, and the matrix production of the tumour, and it is the most important factor to predict overall patient survival and metastatic potential. Patients with well-differentiated tumours (i.e., atypical cartilaginous tumour (ACT) and grade I) have an overall 10-year survival rate of 88–95% and rarely show metastasis formation [1]. However, high-grade tumours (i.e., grade II and III) show increased metastatic potential (10–30% and 32–71%, respectively) and the overall 10-year survival rate of these patients is severely decreased (58–86% and 26–55%, respectively) [2]. Dedifferentiated chondrosarcoma is a high-grade subtype of chondrosarcoma with the bimorphic histological appearance of a conventional chondrosarcoma juxtaposed with a high-grade anaplastic sarcoma [5]. It has a dismal prognosis, with 5-year overall survival of only 7–24%. 

The worse prognosis of both high-grade conventional and dedifferentiated chondrosarcoma can be partially ascribed to the limited number of available treatment options. Chondrosarcomas are intrinsically resistant towards chemo- and radiotherapy and targeted therapeutic options are still lacking, leaving surgery as the only curative treatment option [6]. Hence, there is an urgent need to develop novel targeted therapeutic strategies, especially for patients with metastasised and/or unresectable high-grade or dedifferentiated chondrosarcomas. 

In the last decade, recurrent heterozygous hotspot mutations in the arginine residues of the isocitrate dehydrogenase 1 and −2 (*IDH1* and *IDH2*) genes (p.R132 and p.R140/p.R172, respectively) were identified in enchondroma (87%), central conventional chondrosarcoma (~50%), and dedifferentiated chondrosarcoma (>80%) [7,8,9,10]. The high frequency of *IDH1* and *IDH2* (collectively referred to as *IDH*) mutations in benign cartilage tumours indicates that these mutations occur early in tumourigenesis, suggesting that *IDH* mutations have an important driver role in the formation of cartilage tumours. Indeed, the introduction of an *IDH* mutation induces enchondroma-like lesions in mice [11]. Furthermore, the *IDH* mutation or its produced oncometabolite stimulate chondrogenic differentiation while inhibiting the osteogenic differentiation of mesenchymal stem cells, which are the presumed cells of origin of cartilage tumours [12,13]. Despite their significant role in the early stages of tumour development, the prognostic value of the *IDH* mutation in chondrosarcoma seems controversial and (pre)clinical studies that have focused on the direct and indirect targeting of the *IDH* mutation have not yielded novel treatment strategies. This review provides an overview of the current knowledge of the role of *IDH* mutations in chondrosarcoma and highlights similarities as well as differences between tumour types that frequently harbour *IDH* mutations. Additionally, it will be discussed whether the *IDH* mutation should still be considered as a promising therapeutic target or not. 

## 2. Frequency and Prognostic Value of *IDH1* and *IDH2* Mutations

*IDH* mutations are also frequently observed in other tumour types, such as acute myeloid leukaemia (AML), glioma, and cholangiocarcinoma [14]. Interestingly, the most common variant differs between the above-stated tumour types (Table 1). Cartilage tumours and cholangiocarcinoma mainly have IDH1 p.R132C variants (~60%), glioma predominantly harbours IDH1 p.R132H mutations (~90%), and AML often has IDH2 p.R140Q mutations (~40%) [15,16]. None of the variants are exclusively observed in one tumour type, suggesting that different point mutations can have a similar effect on tumourigenesis, although the level of the oncometabolite D-2-hydroxyglutarate (D-2-HG) produced by these variants differs [17,18,19]. The prognostic value of *IDH* mutations in these tumour types is also diverse (Table 1), and only glioma patients have a clear favourable outcome when their tumour harbours an *IDH* mutation [20,21,22,23]. Studies that were performed to determine the prognostic value of *IDH* mutations in chondrosarcoma show contradictory results. While it was previously reported that *IDH* mutations do not predict outcomes [15], other studies showed either a worse [24] or better [25] prognosis for *IDH* mutant (*IDH*^MUT^) chondrosarcoma patients. The three patient cohorts were similar in size (*n* = 70 to 80) and median age (50 to 60 years), but the chondrosarcoma subtype inclusion (conventional versus addition of dedifferentiated and mesenchymal cases) and median follow-up time (4.3 versus ≥10 years) differed, which might explain the discrepancy in results. Another factor might be the type of technique used to assign patients to the *IDH*^MUT^ subgroup. For instance, Sanger sequencing is not sensitive enough to detect mutations when present in less than <30% of the sequenced PCR product, leading to false-negative results in samples with a low *IDH*^MUT^ variant allele frequency or tumour cell percentage and thereby the assignment of *IDH*^MUT^ patients to the *IDH* wildtype (*IDH*^WT^) subgroup. Despite the lack of prognostic value, the high occurrence rate of *IDH* mutations in all of these tumour types suggests that they have an important role in driving tumourigenesis, already in the early stages of tumour development.

## 3. Oncogenic Activities of *IDH* Mutations

Both IDH enzymes function in the tricarboxylic acid (TCA) cycle, where they convert isocitrate into α-ketoglutarate (α-KG) and CO_2_. Mutated IDH enzymes acquire a neomorphic function, leading to the additional conversion of α-KG into the oncometabolite D-2-HG [39]. The IDH1 p.R132C variant is one of the most efficient D-2-HG producers, while both IDH1 p.R132H and IDH2 p.R140Q produce lower levels of the oncometabolite [17,18,19]. As certain variants are more frequently observed in specific tumour types (Table 1) [15,16], this could suggest that chondrosarcoma and cholangiocarcinoma rely on high D-2-HG levels, while glioma and AML depend on relatively lower levels of the oncometabolite.

Due to the high structural similarity between α-KG and its antagonist D-2-HG, the oncometabolite is able to competitively bind α-KG-dependent enzymes, leading to the overall inhibition of this class of enzymes [40,41]. The inhibition of α-KG-dependent enzymes leads to widespread changes in the epigenomes and metabolomes of cells and affects DNA repair and cellular growth signalling pathways (Figure 1) [42,43]. For instance, the D-2-HG-mediated inhibition of α-KG-dependent DNA demethylases (family of TET enzymes, including TET1/2) and histone demethylases (family of Jumonji enzymes, including KDMA4A/B) leads to an overall DNA hypermethylation phenotype, as well as an aberrant histone methylation phenotype in *IDH* mutant tumours. *IDH*^MUT^ enchondromas and chondrosarcomas are indeed characterised by a CpG island methylator phenotype (CIMP)-positive status, and DNA hypermethylation is present in primary *IDH*^MUT^ chondrosarcomas [7,44,45]. The family of Jumonji enzymes is also involved in the regulation of the Mechanistic Target Of Rapamycin Kinase (mTOR) signalling pathway, as well as DNA repair via the homologous recombination pathway. Moreover, IDH^MUT^ enzymes have a reduced ability to produce NADPH and consume high levels of NADPH to produce D-2-HG, resulting in severely reduced overall NADPH levels. This deficiency does not only cause metabolic stress but will also lead to an increase in reactive oxygen species (ROS), making *IDH*^MUT^ tumours more vulnerable to DNA damage. Besides the induction of metabolic stress, *IDH*^MUT^ tumours also undergo metabolic rewiring, including alterations in metabolites of the TCA cycle, a reduced dependency on glycolysis, and alterations in lipid metabolism. Additionally, D-2-HG-mediated inhibition of the prolyl hydroxylase domain proteins (EGLN1 and -2) leads to the upregulation of hypoxia-inducible factors (e.g., HIF1α), resulting in a metabolic switch to maintain oxygen homeostasis. D-2-HG also affects collagen maturation via the inhibition of proline and lysine hydroxylases (P4HA1-3 and PLOD1-3), leading to an impaired extracellular matrix structure. Thus, *IDH* mutations have a wide variety of downstream biological effects; therefore, these mutations are considered as the drivers in multiple tumour types.

## 4. Inhibition of the IDH^MUT^ Protein

To counteract the oncogenic activity of the *IDH* mutations, several inhibitors targeting either IDH1 p.R132 variants (e.g., ivosidenib) or IDH2 p.R140 variants (e.g., enasidenib) have been developed over the past couple of years [46]. In vitro studies and clinical trials show that AML patients could benefit from IDH^MUT^ protein inhibitors [26,32], although some patients develop resistance against these inhibitors over time. This acquired resistance is multi-factorial and can be caused by second-site mutations in *IDH*^MUT^ genes to prevent the binding of IDH^MUT^ protein inhibitors, *IDH*^MUT^ isoform switching to circumvent the effect of IDH^MUT^ protein inhibitors, or novel acquired mutations in genes encoding for receptor tyrosine kinases (RTKs) [33,34,35]. Direct inhibition of IDH^MUT^ proteins seems less promising for other tumour types that frequently harbour an *IDH* mutation (Table 1) [27,28,36,37]. Especially in chondrosarcoma, the effect of IDH^MUT^ protein inhibitors in in vitro assays seems controversial. While several studies have shown that IDH1^MUT^ protein inhibition does not affect the tumourigenic properties of chondrosarcoma cell lines [27,29], other groups have shown that IDH1^MUT^ protein inhibition causes a decreased proliferation rate in chondrosarcoma cell lines at higher doses or with a different compound [30,31]. Recent results from a phase I clinical trial with the IDH1^MUT^ inhibitor ivosidenib showed that prolonged disease control (i.e., progression-free survival of ~6 months) could be achieved in a subset of patients with advanced chondrosarcoma, predominantly in patients with a minimal number of co-occurring mutations [38]. Together, these results suggest that a subset of chondrosarcomas might have become independent of their *IDH* mutation over time and that the underlying biological changes either have become static or are driven by other mutations that were acquired later during tumour development.

## 5. Synthetic Lethal Interactions with the *IDH* Mutation

As IDH^MUT^ protein inhibitors showed limited efficacy in in vitro assays and clinical trials or acquired resistance was observed (Table 1), a large number of in vitro studies were performed to determine whether directly targeting the downstream biological effects of *IDH* mutations would be more promising (Table 2). Indeed, multiple synthetic lethal interactions with the *IDH* mutation were reported for AML and glioma, including radiotherapy, chemotherapy, and agents that target poly(ADP-ribose) polymerase (PARP), B-cell lymphoma 2 (Bcl-2) family members, Bromodomain and Extra-Terminal Motif (BET) proteins, DNA methyltransferases (DNMTs), mTOR, Nicotinamide Phosphoribosyltransferase (NAMPT), and glutaminase [27,28,47,48,49,50,51,52,53,54,55,56,57,58,59,60]. However, chondrosarcoma cell lines are variably sensitive to a selection of these therapies, but the effect seems irrespective of the *IDH* mutation status, as *IDH*^WT^ chondrosarcoma cell lines show similar treatment responses [61,62,63,64,65,66,67].

These contradictory findings on synthetic lethal interactions with the *IDH* mutation might be ascribed to different factors. First, the cell of origin and the tumour microenvironment (e.g., cartilaginous matrix formation and hypoxia in chondrosarcoma) of the distinct tumour types that frequently harbour an *IDH* mutation are highly different and could therefore influence the role that *IDH* mutations play in tumourigenesis. Second, the level of the D-2-HG oncometabolite may also influence the downstream biological effects of *IDH* mutations. The most common *IDH* variants in AML and glioma both produce relatively low D-2-HG levels, whilst the most common point mutation in both cholangiocarcinoma and chondrosarcoma produces relatively high levels of the oncometabolite (Table 1) [17,18,19]. It was recently shown that a lower level of DNA hypermethylation was observed for the IDH1 p.R132H variant compared to non-p.R132H variants, irrespective of tumour type [16]. Lastly, the type of in vitro model (endogenous vs. artificially created) might influence whether synthetic lethal interactions with the *IDH* mutation are present or not. The introduction of an *IDH* mutation in a glioma model leads to reduced glutamine and glutamate levels, but this change in TCA cycle metabolites is not present when endogenous *IDH*^WT^ and *IDH*^MUT^ glioma models are compared [68]. Most synthetic lethal interactions with the *IDH* mutation were indeed identified in generic cancer cell lines with an introduced *IDH*^MUT^ (Table 2). AML and glioma cell lines with an endogenous *IDH*^MUT^ are scarce, but the utilised chondrosarcoma cell lines do harbour endogenous *IDH* mutations and this difference in model type could explain why synthetic lethal interactions with the *IDH* mutation are absent in the chondrosarcoma in vitro studies. As *IDH* mutations occur early during tumourigenesis, especially in chondrosarcoma, artificial models with an introduced *IDH* mutation may not be representative of the role that *IDH* mutations normally play in tumourigenesis. These studies also introduced the *IDH* mutation in generic cancer cell lines that are easy to transfect (e.g., HeLa, HCT116, and U2OS cells), and these cell lines do not represent the tumour types in which *IDH* mutations frequently occur. Moreover, most studies generated models that overexpressed the IDH^MUT^ protein, whilst the balanced expression of IDH^WT^ and IDH^MUT^ is needed to retain efficient D-2-HG production [69]. Together, these considerations emphasise that the tumour type, the *IDH*^MUT^ variant, and the type of in vitro model should be taken into account when studying synthetic lethal interactions with the *IDH* mutation, and that the underlying vulnerabilities may highly differ between tumour types that frequently harbour an *IDH* mutation.

## 6. Putting the *IDH* Mutation into Context to Define Underlying Vulnerabilities

In addition to these factors, it was recently shown that the (epi)genetic landscape in which *IDH*^MUT^ and *IDH*^WT^ are embedded is another important aspect to take into consideration when defining underlying vulnerabilities in tumour types that frequently harbour an *IDH* mutation. Studies on AML and glioma have shown that the genetic and epigenetic landscape in which *IDH*^WT^ and *IDH*^MUT^ function is highly heterogenous and thereby influences the therapy response and patient outcome [70,71,72,73,74,75,76,77,78,79,80,81]. For instance, mutations in *TP53* and *ATRX* are the underlying denominator in defining which *IDH*^WT^ and *IDH*^MUT^ gliomas respond to radiotherapy [70]; the overexpression of BCAT1 in *IDH*^WT^ AML leads to an *IDH*^MUT^-like DNA hypermethylation phenotype [71], and additional mutations in *DNMT3A* cause reduced levels of DNA hypermethylation in *IDH*^MUT^ AML samples [74]. Furthermore, co-occurring (epi)genetic alterations such as CIMP status [78], 1p19q deletions [80], *CDKN2A* deletions [78,79], *MET* amplifications [78], *PDGFRA* amplifications [79], and *TERT* mutations [80] influence overall survival in *IDH*^MUT^ glioma patients. Moreover, *IDH*^MUT^ AML patients with a co-occurring *NPM1* mutation show overall a better response to chemotherapy with or without venetoclax [81]. The influence of co-occurring (epi)genetic alterations may also explain why distinct *IDH*^MUT^ tumour types differ in therapy sensitivity and underlines the need to use endogenous *IDH*^MUT^ models, as generic cancer cell lines with an introduced *IDH* mutation do not represent the (epi)genetic landscape in which *IDH* mutations naturally exist. Thus, the *IDH* mutation status does not solely define the underlying vulnerabilities, which is in line with previous findings for chondrosarcoma [61,62,63,64,65,66,67], suggesting that a dichotomy between *IDH*^WT^ and *IDH*^MUT^ is too simplistic.

Besides *IDH* mutations, chondrosarcomas frequently harbour mutations in *TP53*, *CDKN2A/B*, *COL2A1*, *YEATS2*, *NRAS*, and *TERT* [82,83,84,85,86]. However, the rest of the previously observed co-occurring mutations seem to follow a more random pattern and are present in less than 10% of the chondrosarcomas [25,83,84,87], leading to a highly heterogeneous genetic landscape in which *IDH*^WT^ and *IDH*^MUT^ function in chondrosarcoma. Furthermore, *IDH*^MUT^ chondrosarcomas are characterised by a global hypermethylation phenotype that changes with increasing histological grade [44,45], and, based on methylation profiles alone, several chondrosarcoma subgroups could be defined, even within *IDH*^WT^ and *IDH*^MUT^ tumours [88]. Moreover, using chondrosarcoma transcriptome and methylome data, it was previously shown that different molecular subtypes (i.e., high mitotic state, 14q32 miRNA cluster loss of expression, and *IDH*^MUT^-induced DNA hypermethylation) exist, and that these are associated with patient outcomes [89]. Moreover, (epi)genetic alterations in the *TERT* gene (i.e., hypermethylation and promotor mutations) affect the survival probability of *IDH1*^MUT^ chondrosarcoma patients, whilst this association is absent in *IDH*^WT^ and *IDH2*^MUT^ patients [87]. Together, these findings show that the *IDH* mutation status does not solely define the treatment response or outcome in chondrosarcoma patients, suggesting that the dichotomy between *IDH*^WT^ and *IDH*^MUT^ is also too simplistic for chondrosarcoma.

## 7. Conclusions and Future Directions

Although *IDH* mutations occur frequently in chondrosarcoma, their prognostic value as well as therapeutic potential seem both ambiguous in chondrosarcoma (Table 1). This is in line with the hypothesis that some chondrosarcomas become independent of their *IDH* mutations over time and that additional mutations take over the driver role in later stages of tumour development. Nevertheless, other tumour types that frequently harbour an *IDH* mutation do show the prognostic value of the *IDH* mutation (glioma) and response to IDH^MUT^ protein inhibitors (AML) (Table 1). Additional mutations in RTKs could contribute to secondary resistance to IDH^MUT^ protein inhibitors [35], which complements the idea that other mutations can take over the driver role of *IDH* mutations. As chondrosarcomas are usually diagnosed relatively late due to minimal symptoms in the early stages of tumour development, these additional mutations might have already occurred and may hamper the efficacy of IDH^MUT^ protein inhibitors. This is in line with the fact that an increase in progression-free survival after treatment with an IDH^MUT^ protein inhibitor (ivosidenib) was predominantly observed in chondrosarcoma patients with a minimal number of co-occurring mutations [38]. Thus, the role of *IDH* mutations most likely differs between distinct tumour types as well as stages of tumour development. Future studies should investigate whether (the number of) additional mutations could be a potential predictive biomarker for the response to IDH^MUT^ inhibition in chondrosarcoma and other *IDH*^MUT^ tumour types.

This discrepancy in the role of *IDH* mutations is also reflected in the preclinical studies that have investigated underlying vulnerabilities in *IDH*^MUT^ tumour types (Table 2). The contradicting results on synthetic lethal interactions with the *IDH* mutation between tumour types might be ascribed to a difference in cell of origin, co-occurring (epi)genetic alterations, D-2-HG levels, or the type of in vitro model. The latter could introduce variance between *IDH*^MUT^ tumour types that does not naturally exist, as these artificially created *IDH*^MUT^ most likely do not reflect the early onset of *IDH* mutations or the (epi)genetic landscape in which these mutations are normally embedded. The endogenous *IDH*^WT^ and *IDH*^MUT^ chondrosarcoma cell lines harbour additional genetic alterations (unpublished data), including *CDKN2A* loss, and were derived from patients with high-grade (II and III) and differentiated tumours, meaning that these cell lines are representative models for the role of *IDH* mutations in more advanced tumour stages. This might explain why these cell lines showed a limited response to IDH^MUT^ inhibitors and showed a variable response to treatments that were identified in other in vitro studies that utilised artificially created models. Future preclinical studies should therefore exercise caution regarding the use of artificially created *IDH*^MUT^ models and should thoroughly characterise the (epi)genetic landscape in which the *IDH* mutation was introduced, as well as confirming their findings in models that harbour an endogenous *IDH* mutation.

Nevertheless, these artificially created *IDH*^MUT^ models could provide valuable insight into the influence of the (epi)genetic landscape on the downstream biological effects of *IDH* mutations and thus on underlying therapeutic vulnerabilities. Future studies should expand on the identified subgroups and define novel (epi)genetic aberrations that distinguish subgroups within *IDH*^WT^ and *IDH*^MUT^ chondrosarcomas, followed by the identification of tailored targeted therapeutic strategies towards these subgroups. This will improve not only the identification of effective treatment modalities but also the design of clinical trials for high-grade chondrosarcoma patients (e.g., umbrella trial design) and the inclusion of patients in basket trials. 

## Figures and Tables

**Figure 1 cancers-15-03603-f001:**
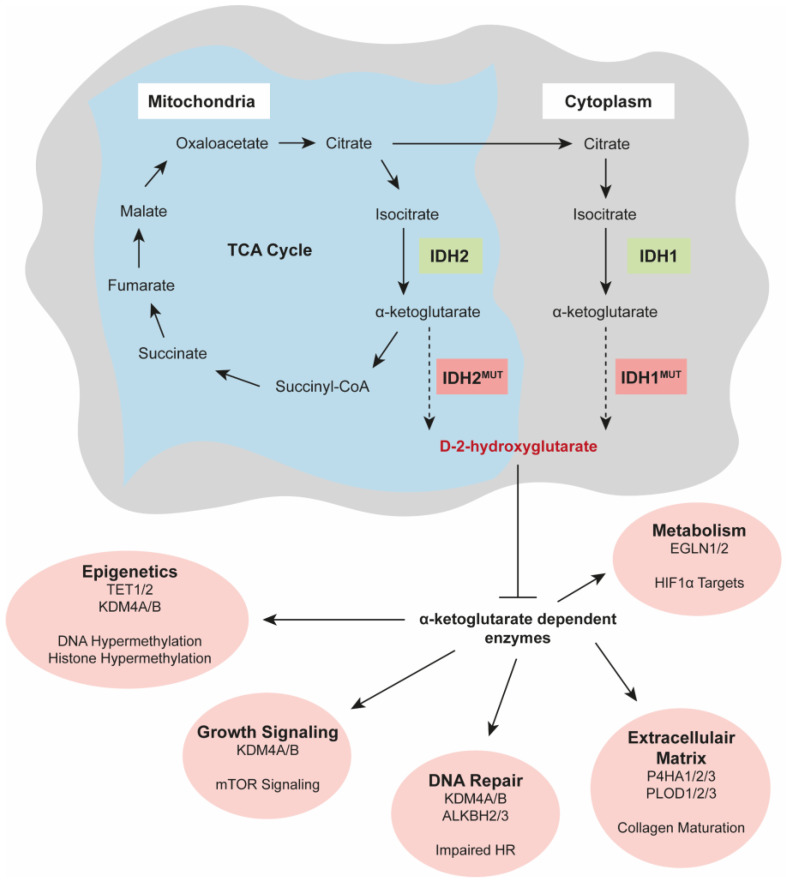
*IDH* mutations have a wide variety of downstream biological effects. Mutated IDH enzymes produce D-2-HG, which is an oncometabolite that competitively binds α-KG-dependent enzymes. Inhibition of this class of enzymes leads to widespread changes in the epigenomes and metabolomes of cells and alterations in DNA repair pathways, extracellular matrix structure, and cellular growth signalling pathways. Besides the inhibition of α-KG-dependent enzymes, *IDH* mutations also cause metabolic stress, metabolic rewiring, the depletion of NADPH, and an increase in ROS.

**Table 1 cancers-15-03603-t001:** *IDH* mutations in different tumour types.

	AML	Glioma	Cholangiocarcinoma	Chondrosarcoma
**Frequency of *IDH*^MUT^**	~10–15% [14]	>70% [14]	~15–20% [14]	~50% [7,8,9]
**Most common IDH^MUT^ variant**	IDH2 p.R140Q (~40%) Weak D-2-HG Producer [16,17]	IDH1 p.R132H (~90%) Weak D-2-HG Producer [16,18]	IDH1 p.R132C (~60%) Strong D-2-HG Producer [16,18]	IDH1 p.R132C (~60%) Strong D-2-HG Producer [15,16,18]
**IDH^MUT^ inhibition** **in vitro**	Differentiation [26]	No effect [27]	No effect [28]	Controversial [27,29,30,31]
**IDH^MUT^ inhibition** **clinical trials**	~40% response, secondary resistance [32,33,34,35]	Less promising, prolonged disease control in subset [36]	Less promising, prolonged disease control in subset [37]	Durable disease control in subset [38]
***IDH*^MUT^ effect on outcome**	No difference (in MDS: worse prognosis) [20,21]	Better prognosis, due to favourable response? [22]	Beneficial? [23]	Controversial [15,24,25]

MDS: myelodysplastic syndrome.

**Table 2 cancers-15-03603-t002:** Synthetic lethal interactions with the *IDH* mutation in in vitro models.

	AML	Glioma	Cholangiocarcinoma	Chondrosarcoma
**Radiotherapy**	Molenaar 2018 [47] *	Li 2013 [48] Kessler 2015 [49]		De Jong 2019 [61] *
**Temozolomide**		Lu 2017 [50] Tateishi 2017 [51] *		Venneker 2019 [66] *
**PARP inh.**	Sulkowski 2017 [52] Molenaar 2018 [47] *	Sulkowski 2017 [52]	Wang 2020 [53]	Venneker 2019 [66] * Palubeckaitė [67] *
**Bcl-2/Bcl-xL inh.**	Chan 2015 [54]	Karpel-Massler 2017 [55]		De Jong 2018 [62] *
**BET inh.**	Chen 2013 [56]	Fujiwara 2018 [28]	Fujiwara 2018 [28] *	
**DNMT inh.**		Turcan 2013 [57] *		
**mTOR inh.**		Batsios 2019 [58]		Addie 2019 [63] *
**NAMPT inh.**		Tateishi 2015 [27] *		Peterse 2017 [64] *
**Glutaminolysis inh.**	Emadi 2014 [59] *	Seltzer 2010 [60]		Peterse 2018 [65] *

inh.: inhibitors, * endogenous *IDH*^MUT^ models. Green: Sensitive; Red: Not sensitive; Yellow: Sensitive, but irrespective of *IDH*^MUT^.

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
