# Peer review of "IDH Mutations in Chondrosarcoma: Case Closed or Not?"

_cancers, 2023, doi:10.3390/cancers15143603_

Round 1
Reviewer 1 Report
In this comprehensive and well structured review authors discussed IDH1 and IDH 2 mutations regarding their presence and frequencies in AML, glioma, cholangiocarcinoma and chondrosarcomas. In addition, the different oncogenic activities of IDH mutations were illustrated and the effect of IDHMUT inhibitors in clinical trials reviewed. Finally, preclinical approaches using synthetic lethal interactions with the IDH mutation in in vitro models of the four different malignancies were discussed and suggestions for future directions in order to better define vulnerabilities in IDHMUT malignancies were provided.
The critical comparison of results obtained in the different models, artificially created versus endogenous IDHMUT cell lines, was very much appreciated.
Minor comments
1) page 3, table 1, second column:
The abbreviation "MDS" should be explained in a footnote.
2) page 3, lines 104 - 108
Since three studies have reported contradictory results for chondrosarcoma patients, it would be interesting if authors could comment on possible reasons, like number and type of patients (localised, ...), technical issues, ethnical differenes, treatment strategies and others.
3) page 6, table 2, first column:
The abbreviation "inhib." should be explained or substituted with a simbol and explained in a footnote that it stands for inhibitors.
Reviewer 2 Report
- This is a review article on the role of IDH mutations in the pathophysiology of chondrosarcoma and the potential of IDH mutations as a biomarker and treatment target submitted by-well established researchers in the field. The authors succinctly review the disparate results of these aspects of IDH mutations in four different tumor types including chondrosarcoma. They then propose hypotheses to explain the discrepant results and future directions for research to unravel this complex biology. The paper is certainly worthy of publication in a Special Issue on Latest Research in Cartilaginous Neoplasms. I do not have any recommendations for any improvements. It is very good and thorough as is.
Line 75, are there genetic models with lineage speceific IDH mutations that result in enchodmomas?
Line 14: delets "exist"
Line 55: a.o. analysis of?
Line 86 (in)direct, do you mean direct and indirect?
Refs 1-4 could be combined.
Reviewer 3 Report
This review paper is well written and thorough. My only recommendation would be to include photomicrographs of the histology of the different grades of chondosarcomas. This may make it easier for the reader to understand what the author have written about these. It is not required but recommended change.
